# The importance of degradation mode analysis in parameterising lifetime prediction models of lithium-ion battery degradation

Ruihe Li[1,2], Niall D. Kirkaldy [1,2], Fabian F. Oehler[3], Monica Marinescu[1,2], Gregory J. Offer [1,2] ✉ & Simon E. J. O'Kane[1,2]

Predicting lithium-ion battery lifetime remains a critical and challenging issue in battery research right now. Recent years have witnessed a surge in lifetime prediction papers using physics-based, empirical, or data-driven models, most of which have been validated against the remaining capacity (capacity fade) and sometimes resistance (power fade). However, there are many different combinations of degradation mechanisms in lithium-ion batteries that can result in the same patterns of capacity and power fade, making it impossible to find a unique validated solution. Experimentally, degradation mode analysis involving measuring the loss of lithium inventory, loss of active material at both electrodes, and electrode drift/slippage has emerged as a state-of-the-art requirement for cell degradation studies. This work represents the integration of five distinct degradation mechanisms. We show how three models with different levels of complexity can all fit the remaining capacity and resistance well, but only the model with five coupled degradation mechanisms could also fit the degradation modes at three temperatures. This work proves that parameterizing using only capacity and power fade is no longer sufficient, and experimental and modelling degradation studies should include degradation mode analysis for parameterization in the future.

Due to the requirements in electric vehicles, smart phone and energy storage stations, the demand of lithium-ion batteries (LIBs) is expected to increase by 33% each year from 2022 to reach ~ 4700 GWh by 2030[1]. The performance of LIBs degrades with time and repeated cycling[2]. The production and recycling of LIBs poses huge environmental and financial challenges to the whole society[3], making degradation of LIBs a big concern.

To understand the degradation behaviours of LIBs, a computational model is required. Among different types of models, physics-based models are preferred for identifying degradation mechanisms and their effects on battery performance because they can account for the root causes of degradation. Examples of the usage of physics-based degradation models include predicting remaining useful life (RUL)[4], optimising operation conditions[5] and improving manufacturing processes[6]. The insights from physics-based models can also feed into empirical models and data-driven models to benefit from the ease of use of these models and reduce computational time[7].

To achieve the above benefits, the model must be well-parameterised and validated against experimental measurements. The gold standard for the last decade has been to reproduce multiple ageing

[1]Department of Mechanical Engineering, Imperial College London, London SW7 2AZ, UK. [2]The Faraday Institution, Quad One, Becquerel Avenue, Harwell Campus, Didcot, OX11 0RA, UK. [3]Technical University of Munich (TUM), Institute for Electrical Energy Storage Technology (EES), Arcisstr. 21, 80333 Munich, Germany. ✉e-mail: gregory.offer@imperial.ac.uk

features, normally in the form of capacity retention curves, under as many ageing conditions as possible. For example, the square root of time dependency can be reproduced by a diffusion-limited process[8]. The temperature dependency can be depicted by two Arrhenius relationships: one for high temperatures[8,9], and one for low temperatures, with an optimum operating temperature in the middle, at around 25 °C[10,11]. Capacity recovery during the early stage of ageing can be captured by considering anode overhang[12]. Rollover failure can be explained by SEI coupling with lithium plating[13], particle cracking coupling with SEI on cracks[11], or SEI cracking coupling with electrode dry-out[14].

In recent years, the importance of coupling together different ageing mechanisms was emphasised[11,15–17]. However, the number of fitting parameters in these coupled models can exceed 10. Many of them are not yet possible to measure with classical electrochemical tests. Instead, they can only be obtained through directly fitting the ageing data. In that case, overfitting becomes a big concern. Most previous papers have validated their models against capacity retention, which is the simplest and most easily measured performance index. There are a few published works that validate against resistance[18,19], full cell dQ/dV[20,21], and SEI thickness[22]. More recently, degradation mode (DM) analysis has been used to link degradation mechanisms and degradation effects[23]. This work focuses on three DMs: loss of lithium inventory (LLI), loss of active material in the negative electrode (LAM$_{NE}$) and in the positive electrode (LAM$_{PE}$). Baure and Dubarry found that the LAM$_{NE}$:LLI ratio can be an effective index to identify accelerated degradation[24]. Therefore, DMs have been used as both parametrization and validation indices in empirical ageing models[23]. However, to the best of our knowledge, DMs have never been used as validation indices in physics-based degradation models. In this work, we highlight the importance of

DMs in parameterising lifetime prediction degradation models. We first propose three different degradation models based on the different combinations of five degradation mechanisms and parameterise them with an ageing dataset. We then compare the abilities of these three models in fitting the different indices of the ageing dataset, which include not only the commonly used indices of voltage, capacity, and resistance but also DMs. Finally, we validate all three models with another ageing dataset and discuss how different aspects of experimental observations can be reproduced by different degradation mechanisms and parameters, which facilitate a comprehensive assessment of the complicated degradation models.

## Results
### Ageing mechanisms
Five degradation mechanisms are considered in this work (Fig. 1): (1) SEI growth, (2) electrolyte dry-out, (3) lithium plating, (4) loss of active material and (5) particle cracking due to mechanical stress. These five degradation mechanisms depicted in Fig. 1 are closely coupled together. To start with, SEI layer growth consumes useable lithium, reducing lithium inventory (increasing LLI), and solvent, potentially leading to electrolyte drying[25]. The electrolyte drying, in turn, renders part of the active surface area inaccessible to intercalation reactions[14]. Mechanical loss of active material from the electrodes reduces the active surface area[11]. As a result, the remaining active surface area sustains higher interfacial current density under applied current, thus accelerating SEI growth, particle cracking, and mechanically induced LAM[26]. Particle cracking exposes fresh electrode surface area to the electrolyte, which triggers rapid SEI layer growth[11,27]. The increased interfacial current density also accelerates lithium plating[28–30]. The

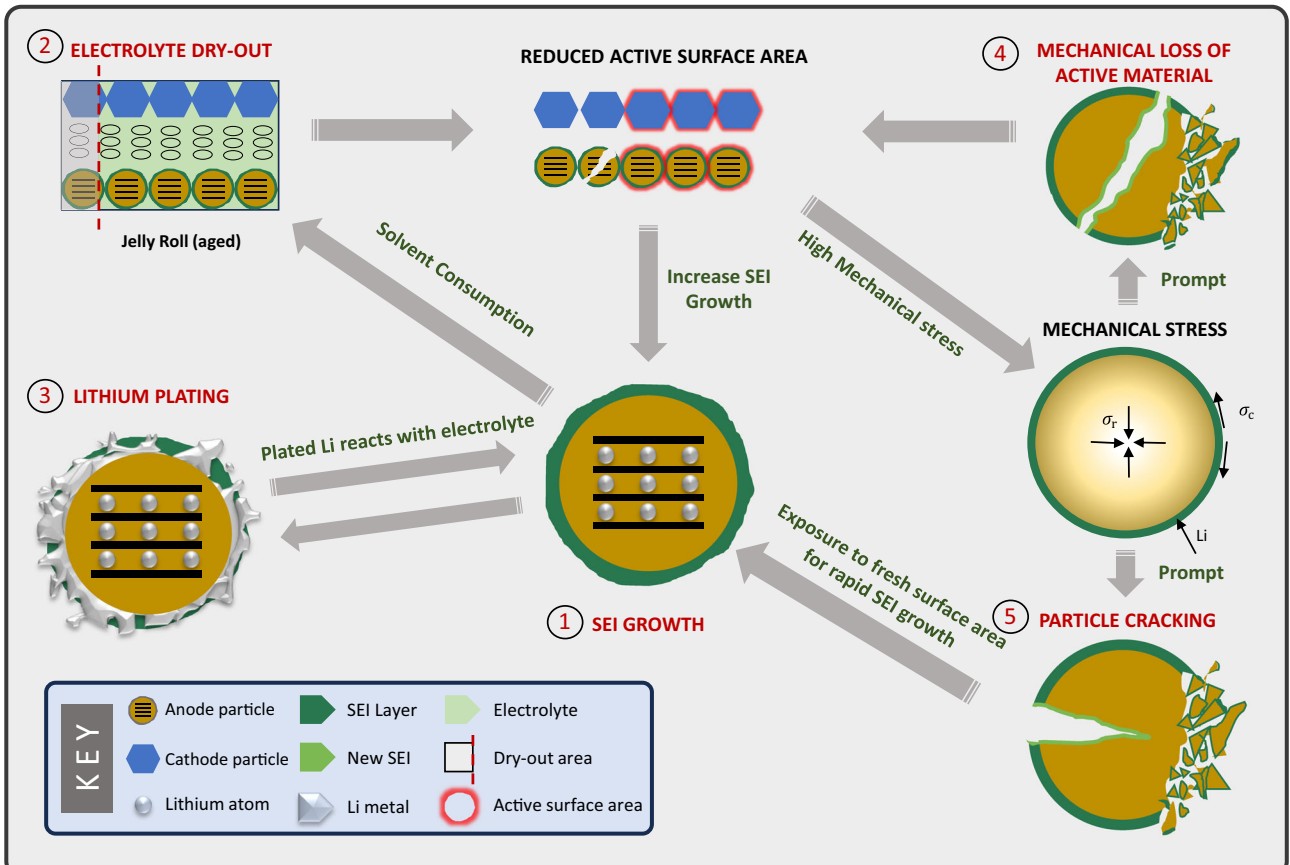

**Fig. 1 | Overview of the degradation model used in this work.** The degradation model includes five degradation sub-models, labelled 1 to 5. They are (1) SEI growth, (2) electrolyte dry-out, (3) lithium plating, (4) loss of active material and (5) particle cracking due to mechanical stress. Their interactions are explained comprehensively in the **Ageing Mechanisms** section.

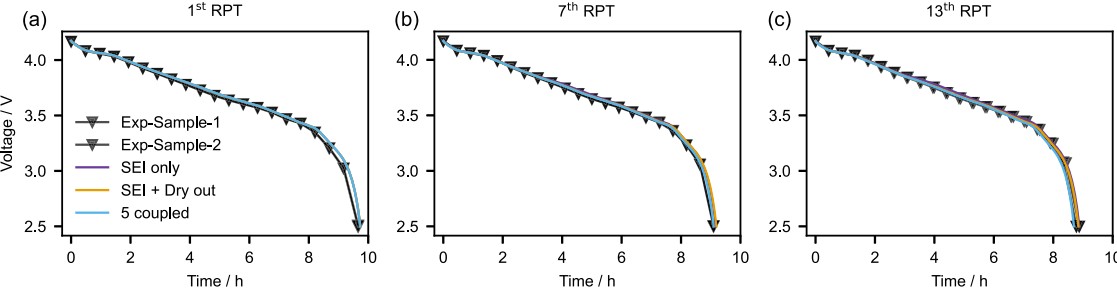

**Fig. 2 | Comparison of three models with the data of Experiment 2 on voltage curves.** These are C/10 discharge voltage curves at 25 °C for the 1st (**a**), 7th (**b**), and 13th (**c**) reference performance tests (RPT). In each plot, the two grey lines represent the two test cells, which align with each other. Source data for this figure are provided as a Source Data file.

plated lithium can itself react with electrolytes form SEI and become completely inactive[11].

It must be noted that not all mechanisms play a significant role in limiting battery performance when the cell is subjected to a specific operating condition. Therefore, not all mechanisms are typically needed to fit one specific ageing dataset. Following the principle of Occam's razor[31], we have picked one, two, and five mechanisms, respectively, to make three degradation models, ranging from simple to complicated. They are: (1) SEI only; (2) SEI with solvent consumption, denoted as SEI + Dry out; (3) all five mechanisms (as included in Fig. 1), further denoted as 5 coupled. SEI layer growth can be considered as the most common degradation mechanism occurring for a wide range of relevant current and temperature conditions and one used in almost all previous physics-based degradation modelling papers. For this reason, it is included in all three sub-models studied. However, the SEI layer growth alone cannot produce LAM, which is observed in the experimental data. To reflect LAM in a simple way, the electrolyte dry-out mechanisms, an intrinsic side effect of SEI layer growth, is added to make the second model. Finally, the third model with all five mechanisms is chosen to test whether an extremely complicated model can outperform the simpler ones in any meaningful ways.

## Simulation errors for voltage, capacity and resistance

Similar to most previous modelling papers, we first compare the simulation errors of the three degradation models for voltage, capacity, and resistance in Experiment 2 (the name of the experiments follows the original publication[32], see Supplementary Note 1). Figure 2 shows the C/10 discharge voltage of the experiment and the three models aged at 25 °C at the first, 7th, and 13th RPT; the discharge voltage curves at 10 °C and 40 °C are presented in Supplementary Fig. 1. In Fig. 2, the three models show excellent agreement with each other and the experimental data. The excellent agreement is further confirmed by the mean percentage errors (MPEs, Table 1 and Supplementary

Fig. 2) and root mean square error (RMSE, Supplementary Fig. 2 and Supplementary Table 1). To be specific, the average MPEs of all RPTs are all below 0.8% (Table 1), and the average RMSE are all below 40 mV for the 3 models, indicating very good fits for the voltage. Therefore, the 3 models all perform well in reproducing the voltage. The 5 coupled model is slightly better than the SEI + Dry out model, and the SEI + Dry out model is slightly better than the SEI-only model.

Figure 3 shows the post-processed experimental data from RPTs for the state of health (SOH, upper column) and 0.1 s resistance (lower column) for three temperatures, alongside predictions from the three models considered. SOH is defined as the C/10 capacity during each RPT over that of the first RPT. The 0.1 s resistance is obtained by the voltage drop of 0.1 s after the 12th pulse of a C/2 GITT discharge[32]. The two light grey lines correspond to the two tested cells, while the black line represents their average. The three coloured lines correspond to the three models. The MPEs of SOH and 0.1 s resistance of the three models is presented in Table 2. Based on Fig. 3 and Table 2, the SEI only and SEI + Dry out model fit SOH at 25 °C relatively well, with MPEs of 0.15 and 0.32, respectively. However, both overestimate SOH at 10 °C and 40 °C. The 5 coupled model, by contrast, achieves a lower simulation error in SOH at 10 °C and 40 °C, with MPEs of 0.42 and 0.57, but underestimates SOH at 25 °C. The 0.1 s resistance predicted by the three models has the same trend as the experimentally obtained resistance at 25 °C, but the resistances predicted for 10 °C and 40 °C are higher than those obtained experimentally. Overall, all three models fit SOH well, with a maximum MPE under 0.88%. The simulation results on the 0.1 s resistance show a higher error. However, the three models yield similar values for the resistance MPEs. No preferred model can be chosen based on the simulation performance of SOH and 0.1 s resistance.

## Simulation error for degradation modes (DMs)

For a more thorough comparison between the three models, we compare their fits against experimentally obtained DMs, as illustrated in Fig. 4 (Calculations of DMs can be found in Supplementary Note 12). The three models all evolve LLI and thus fit the experimentally obtained LLI with different accuracies. While two of the models do predict LAM increase, the SEI-only model, by definition, predicts zero LAM in both electrodes, which leads to 100% MPEs.

For all three temperatures, the SEI-only model significantly underestimates LLI and, therefore, has the highest MPE_LLI (Table 2). However, this simple model still achieves an excellent fit to SOH, with a MPE_SOH of 0.15 at 25 °C. These findings appear to contradict each other until Fig. 3a–c is inspected closely. The SEI-only model predicts that both SOH and LLI follow a square root of time dependence, whereas the SEI + Dry out model predicts a trajectory somewhere between square root and linear behaviour. The experimental SOH measurements have elements of both, which means both models can fit the SOH.

The 5 coupled models fit all DMs better than the SEI + Dry out model at all three temperatures (Table 2). The improvement is significant for low temperatures compared to higher temperatures, as the mechanical

## Table 1 | Average values and standard derivations of MPEs of the C/10 voltage curves for all the 13 RPT cycles in Experiment 2

| Model | T / °C | Mean MPE / % | Standard derivation of MPE / % |
|---|---|---|---|
| SEI only | 10 | 0.72 | 0.08 |
| SEI only | 25 | 0.70 | 0.08 |
| SEI only | 40 | 0.80 | 0.07 |
| SEI + Dry out | 10 | 0.58 | 0.11 |
| SEI + Dry out | 25 | 0.32 | 0.18 |
| SEI + Dry out | 40 | 0.63 | 0.18 |
| 5 coupled | 10 | 0.44 | 0.13 |
| 5 coupled | 25 | 0.48 | 0.18 |
| 5 coupled | 40 | 0.52 | 0.20 |

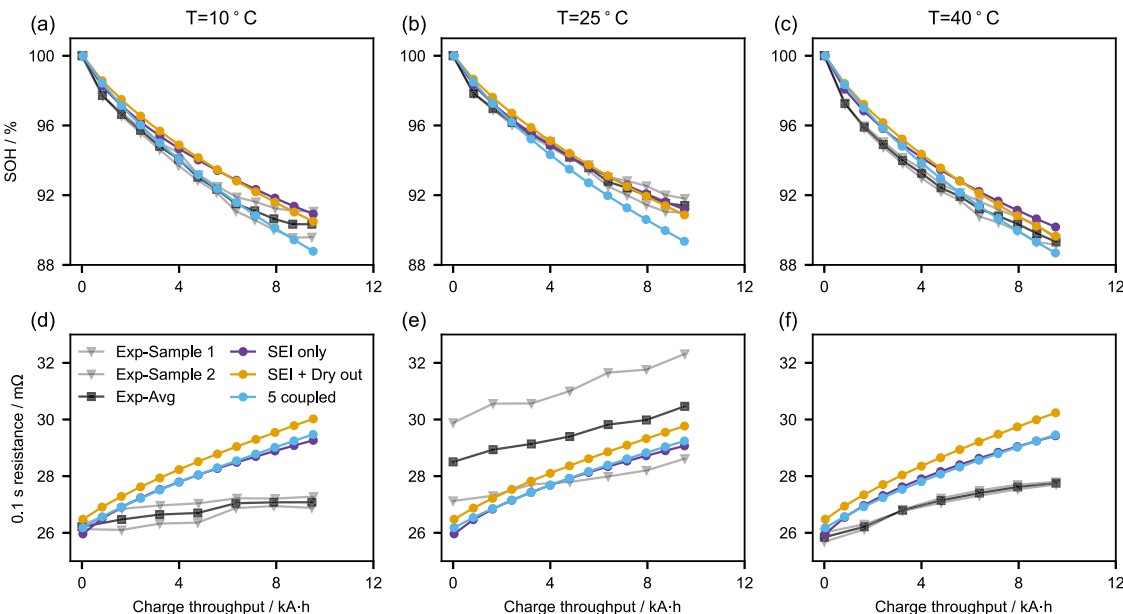

**Fig. 3 | Comparison of three models with the data of Experiment 2 on state of health (SOH) and 0.1 s resistance. a–c** are the SOH, and **d–f** are 0.1 s resistance of the 3 models at 3 temperatures. The grey lines represent the two test cells, and the black line is their average. Source data for this figure are provided as a Source Data file.

**Table 2 | MPEs for all degradation modes, models, and temperatures for Experiment 2. The MPE$_{tot}$ column is a weighted index defined in Eq. (1)**

| Model | T / °C | MPE$_{SOH}$ / % | MPE$_{Res}$ / % | MPE$_{LLI}$ / % | MPE$_{LAM_{NE}}$ / % | MPE$_{LAM_{PE}}$ / % | MPE$_{tot}$ / % |
|---|---|---|---|---|---|---|---|
| SEI only | 10 | 0.83 | 4.41 | 44.45 | 100.00 | 100.00 | 31.52 |
| SEI only | 25 | 0.15 | 5.74 | 41.69 | 100.00 | 100.00 | 31.00 |
| SEI only | 40 | 0.86 | 3.77 | 45.86 | 100.00 | 100.00 | 31.63 |
| SEI + Dry out | 10 | 0.87 | 6.13 | 36.25 | 53.57 | 35.88 | 16.91 |
| SEI + Dry out | 25 | 0.32 | 4.05 | 33.84 | 38.17 | 32.89 | 13.78 |
| SEI + Dry out | 40 | 0.86 | 5.77 | 37.56 | 44.36 | 24.45 | 14.45 |
| 5 coupled | 10 | 0.42 | 4.49 | 25.10 | 37.64 | 27.02 | 11.99 |
| 5 coupled | 25 | 0.88 | 5.50 | 22.75 | 20.58 | 30.94 | 10.41 |
| 5 coupled | 40 | 0.57 | 3.68 | 31.58 | 34.18 | 24.64 | 12.05 |

degradation in the 5 coupled model is more severe at low temperatures. This is because low temperatures lead to lower solid diffusivities of the electrodes (Supplementary Eq. S18) and, therefore, larger lithium concentration gradients in the particles and ultimately more stress-driven LAM (Supplementary Eq. (S20) and Supplementary Fig. 3). Regarding the overall judgement, the 5 coupled model has lower overall MPE$_{tot}$ for the three temperatures (11.99% + 10.41% + 12.05 = 34.45%) compared to that of the SEI + Dry out model (16.91% + 13.78% + 14.45% = 45.14%). The SEI-only model has the highest MPE$_{tot}$ due to the 100% MPEs in LAM for both electrodes. The performance of the three models is initially difficult to be distinguished under voltage, SOH, and 0.1 s resistance but now quite clear using DMs. This highlights the importance of including detailed DMs when parameterising degradation models.

In both the simulations and experiments, the DMs are calculated by tracking how the full-cell pseudo-OCPs change over time, relative to the half-cell OCPs from literature. Modelling provides the additional capability to track how the simulated half-cell pseudo-OCPs themselves change. One powerful way to visualise how the pseudo-OCPs change is to use differential voltage analysis (DVA)[33]. For the reader unfamiliar with these curves, Weng et al.[34] provide a good overview and description of how they change in response to different degradation modes. Figure 5 shows simulated DVA curves for C/10 discharges of the cell aged at 10 °C. Results for 25 °C and 40 °C are presented in Supplementary Fig. 4.

Comparing the simulations at BOL and end of life, the SEI-only model leads to a shift of the half-cell potential curves in relation to each other, which can be recognised in the DVA in Fig. 5c by the leftward shift of the positive electrode potential. This shift is caused by LLI making the highly lithiated, low potential state of the positive electrode inaccessible. However, the shapes of the half-cell curves are unchanged, as demonstrated in Fig. 5b, where the BOL and SEI-only DVA curves are on top of each other. This is a direct result of the SEI-only model being unable to predict any LAM.

For the SEI + Dry out and 5 coupled models, the DVA curves do change shape between BOL and end of life, due to the LAM predicted by those models. The negative electrode pseudo-OCPs for those models are squeezed due to a smaller amount of active material being cycled over the same potential range, resulting in the peaks of corresponding DVA curves in Fig. 5b being larger and in different positions. The positive electrode DVA curves in Fig. 5c behave the same way, except the leftward shift due to LLI also occurs. The simulated full-cell OCPs in Fig. 5a show the combination of all three effects: reduced stoichiometry windows due to LLI, and sharper peaks resulting from LAM in both electrodes.

**Model validation**
The validation of three models is carried out using Experiment 3. The comparison of SOH, 0.1 s resistance and DMs between model and experiment is presented in Fig. 6. Comprehensive evaluations, such as

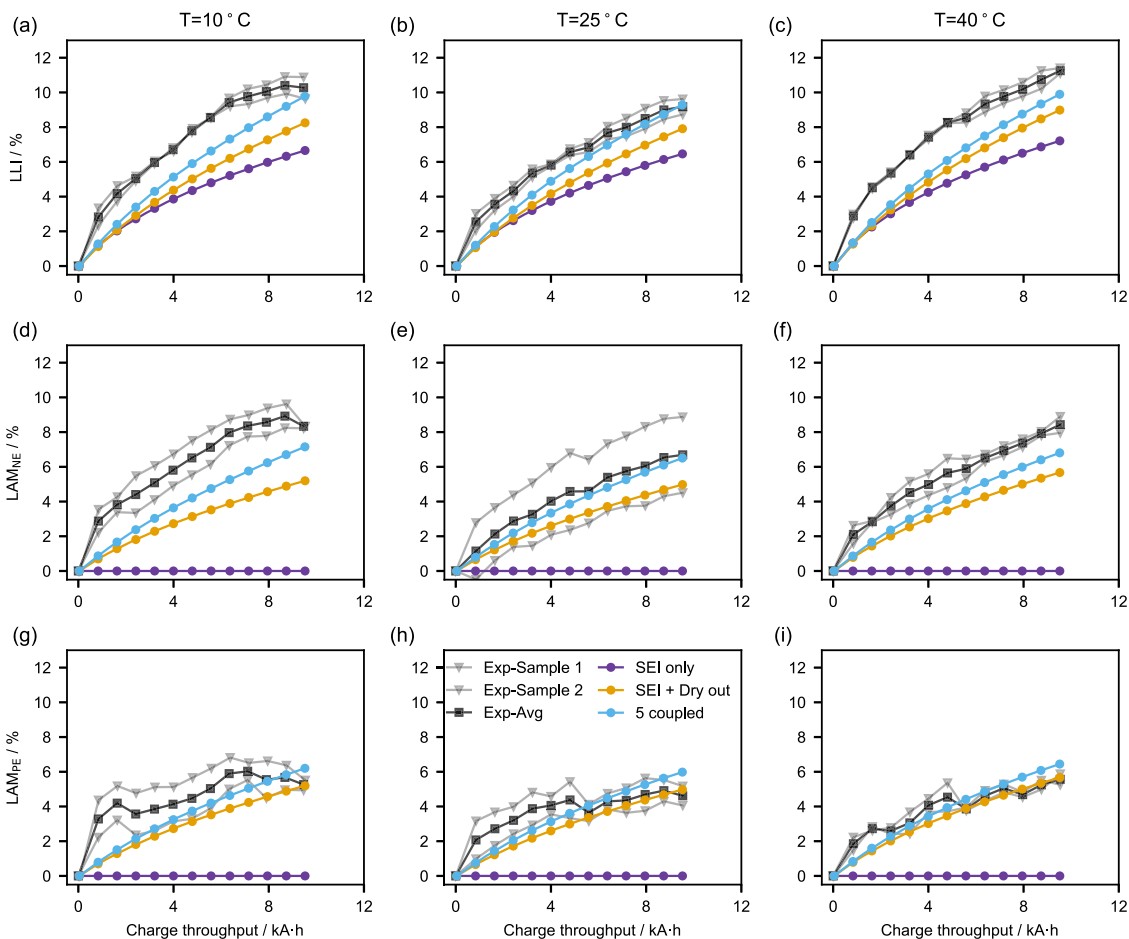

**Fig. 4 | Comparison of three models with the data of Experiment 2 on DMs. a–c** are the loss of lithium inventory (LLI), (**d–f**) are the loss of active material negative electrode (LAM$_{NE}$), and (**g–i**) are the loss of active material positive electrode (LAM$_{PE}$) of the 3 models at 3 temperatures. The grey lines represent the two test cells, and the black line is their average. Source data for this figure are provided as a Source Data file.

MPEs and RMSEs, can be found in Supplementary Tables 10–12 and Supplementary Figs. 14–16. The 5 coupled model well predicts all the five indices of Experiment 3 at 40 °C, leading to an MPE$_{tot}$ of 10.58%. It only deviates from the experiment data at the last RPT, where a small jump in SOH occurs, which can be ascribed to the changes in LAM$_{NE}$ and LAM$_{PE}$. The performance of the 5 coupled model at 25 °C is worse than that at 40 °C but still predicts comparable values at the last RPT, while it completely failed at 10 °C. This may be due to the relatively poor quality of the experiment data, as evidenced by the capacity recovery observed at 10 °C and 25 °C and the huge discrepancy between the three curves at 10 °C, indicating the occurrence of degradation mechanisms that may not have been included in this work. Nonetheless, the 5 coupled model performs the best among the three models in capturing the degradation behaviours observed during cycling at high SOC ranges (85% - 100% in Experiment 3), whereas the other two models underestimate the degradation in all indices in Fig. 6. The best performance of 5 coupled model on Experiment 3 can be ascribed to its ability to predict extra stress driven LAM$_{NE}$ at high SOC range (Supplementary Fig. 5).

**Temperature and SOC dependent degradation**
Other than the indices in one specific ageing condition (under one ageing temperate and one SOC range) investigated above, it is important for degradation models to retrieve other aspects observed when comparing different ageing conditions, such as temperature and SOC dependency.

In Experiment 2, the capacity loss vs temperature is expected to have an asymmetric V-shaped curve. To retrieve this shape in all three

models, we have tuned $E_{act}^{SEI}$ and $E_{act}^{D_{s,n}}$ during parameterisation (Supplementary Fig. 13). $E_{act}^{SEI}$ and $E_{act}^{D_{s,n}}$ are respectively originated from the thermally activated kinetics of SEI reaction and Li diffusion processes. These two parameters have a competing effect on the rate of SEI reaction, in that: on one hand, the higher the $E_{act}^{SEI}$, the lower the SEI reaction at lower temperature; while on the other hand, the higher the $E_{act}^{D_{s,n}}$, the lower the diffusivity and the lower the negative electrode potential during charge, ultimately leading to the higher SEI growth rate. These competing effects make the temperature dependence of SEI growth more complex than is commonly assumed in the literature. Because dry out is a direct result of the SEI growth, the SEI only and SEI + Dry out models share the same temperature dependency. The same combination of $(1 \times 10^4, 6 \times 10^4)$ J/mol for $(E_{act}^{SEI}, E_{act}^{D_{s,n}})$ have been identified for these two models. However, because the SEI + Dry out model has extra capacity loss due to dry out, its inner SEI lithium interstitial diffusivity ($D_{int}$) is about half of the value for the SEI-only model. For the 5 coupled model, due to the contributions of other mechanisms, $D_{int}$ is even smaller. Apart from the same temperature dependency of $E_{act}^{SEI}$ and $E_{act}^{D_{s,n}}$ on SEI and dry-out, $E_{act}^{D_{s,n}}$ also affects the stress-driven LAM$_{NE}$, i.e., a higher value of $E_{act}^{D_{s,n}}$ gives a lower solid diffusivity under low temperatures and, therefore, a higher stress-driven LAM. As a result, a combination of $(5 \times 10^3, 2 \times 10^4)$ J/mol for $(E_{act}^{SEI}, E_{act}^{D_{s,n}})$ have been identified for the 5 coupled model.

However, the temperature dependency of Experiment 3 is different from that of Experiment 2, manifesting more capacity loss in low temperatures. In Supplementary Fig. 6a, the capacity loss (represented by SOH in this work) of Experiment 3 at 10 °C is much larger than that of

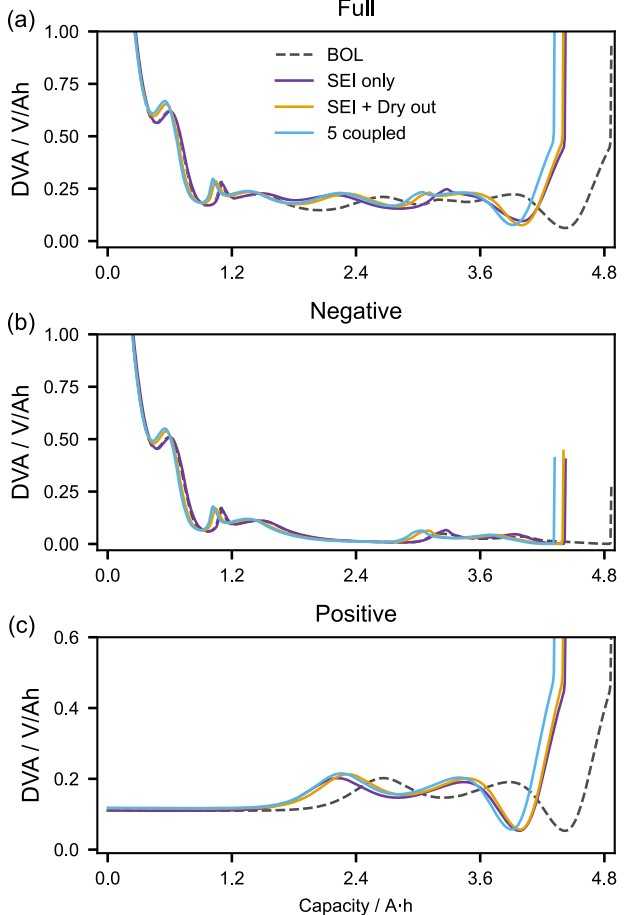

**Fig. 5 | Simulation results of the differential voltage analysis (DVA) at 10 °C.** Results for the full cell (**a**), negative electrode (**b**), and positive electrode (**c**) are included. In each plot, the dashed line corresponds to the beginning of life (BOL), and the three solid lines for the 13[th] RPT for the three different models aged at 10 °C using the protocols of Experiment 2. Source data for this figure are provided as a Source Data file.

Experiment 2 at 10 °C; whereas at 40 °C, the capacity loss between the two experiments is smaller. This may indicate one degradation mechanism is significantly promoted at higher SOC ranges and low temperatures. It is possible that this mechanism is within the five degradation mechanisms considered in this work (such as lithium plating), but corresponding parameters are not tuned properly to manifest such an effect, or it may be that this mechanism is not considered in this work. Nevertheless, we highlight the 5 coupled models still capture the SOC dependency between Experiment 2 and Experiment 3 at 40 °C.

## Discussion

As pointed out in Edge et al.[2], battery degradation can be described with three tiers of detail (Fig. 9 in ref. [2]). The degradation mechanisms are the actual physical and chemical processes occurring inside the battery and thus most fundamental and difficult to measure. Conversely, the easy-to-measure, directly observable effects such as capacity fade, and resistance increase are the cumulative effect of the overall degradation mechanisms and, therefore, deliver limited information, both for diagnosis and prognosis of a meaningful state of health. The DMs are an informative intermediate layer, grouping the effect of different degradation mechanisms based on their overall impact on the cell's thermodynamic and kinetic behaviour. DMs can be obtained from low-rate cycling of the cell and the tear-down half-cell OCP, making them more practical to measure, albeit at a cost, than degradation mechanisms. For example, although the dry-out sub-

model describes the evolution of LAM, the modelled mechanism implicitly predicts $LAM_{NE} = LAM_{PE}$, which, to our best knowledge, is not observed in previous experiment works[32,35–37]. Thus, degradation mechanisms which induce different amounts of LAM in the two electrodes (in our case, mechanical LAM) must be included. Overall, we have found that including DMs as an intrinsic part in model parametrisation is beneficial in model selection and improving model accuracy. For this reason, we propose that any trustworthy model for degradation prediction must include the relationships between degradation mechanisms and degradation modes.

Despite the advancement, we acknowledge that there are some limitations in this work. Firstly, no optimisation methods have been used in parameterisation, which means that there may be other parameter combinations that can fit the experimental data better. Secondly, all three models are unable to retrieve the complicated SOC dependency observed between Experiment 2 and Experiment 3 at 10 °C and 25 °C. Thirdly, all three models are unable to capture Experiment 1 and Experiment 5 in Kirkaldy et al.[32] (see Supplementary Fig. 7). We have not tried Experiment 4 due to the long computational time as it involves a drive cycle. Notably, all three experiments involve cycling the cells at low SOC, where the mechanical degradation of silicon is required[38]. In the future, more ageing mechanisms, such as silicon degradation, will be included to allow the validation against more experiments and more cells, leading to a general enough degradation model.

## Methods
### Models
The Doyle-Fuller-Newman (DFN) pseudo-2D model[39] of LIBs is chosen for representing the beginning of life (BOL) behaviour of the battery. A lumped thermal model[40] is included to capture the temperature-dependent ageing behaviour. For the five aging sub-models, the electrolyte dry-out sub-model is based on Li et al.[25], the SEI growth model is based on Single et al.[41] and von Kolzenberg et al.[42]. The ageing sub-models of lithium plating, loss of active material and particle cracking due to mechanical stress are from O'Kane et al.[11]. Details of the model equations can be found in the original papers, Supplementary Tables 2–4 and Supplementary Notes 2–6. Supplementary Fig. 8 is an illustration of the different types of volumes in the dry-out model. A modified version of PyBaMM's zero-state hysteresis model was used in this work; the equations are listed in Supplementary Note 7, and validation presented in Supplementary Fig. 9.

### Ageing datasets
Two ageing datasets from Kirkaldy et al.[32] are used in this work, one for parametrisation and one for validation. The ageing datasets are obtained from the commercial 21700 cylindrical cells (LGM50T), which have a $SiO_x$-doped graphite negative electrode alongside an NMC811 positive electrode, and a nominal 1 C capacity of 5 Ah. The two ageing datasets correspond to Experiment 2 and Experiment 3 in ref. [32]. In Experiment 2, the cells are cycle aged under 3 different temperatures (10, 25 and 40 °C) between 70% - 85% state of charge (SOC); in Experiment 3, the cells cycled under the same temperature as Experiment 2 but with a SOC range of 85%–100%. The temperatures of cells are controlled using base-cooling. Note that they have also carried out another three-cycle ageing experiments at other SOC ranges, which the models in this work have not been unable to fit yet. The information about the five experiments and the reference performance test (RPT) are briefly summarised in Supplementary Table 5 and Supplementary Note 1. For more comprehensive information, the readers are referred to Kirkaldy et al.[32]

### Parametrisation
Parameters needed in this work can be divided into BOL parameters used in the DFN model and lumped thermal model, and ageing parameters used in the ageing sub-models. The BOL parameters for the LG M50 cell are from O'Regan et al.[40], which features concentration-dependent

diffusivities in the electrode particles and temperature dependency of many parameters that were not included in O'Kane et al.'s[11] model. Dr. O'Regan also performed updated measurements of the half-cell open-circuit potentials (OCPs), which are consistent with their group's earlier measurements from Chen et al.[43] but extend over larger lithiation ranges, improving the accuracy of the degradation mode analysis. Details of these changes can be found in Supplementary Figs. 10–12, Supplementary Tables 6 and 7, and Supplementary Notes 8–11.

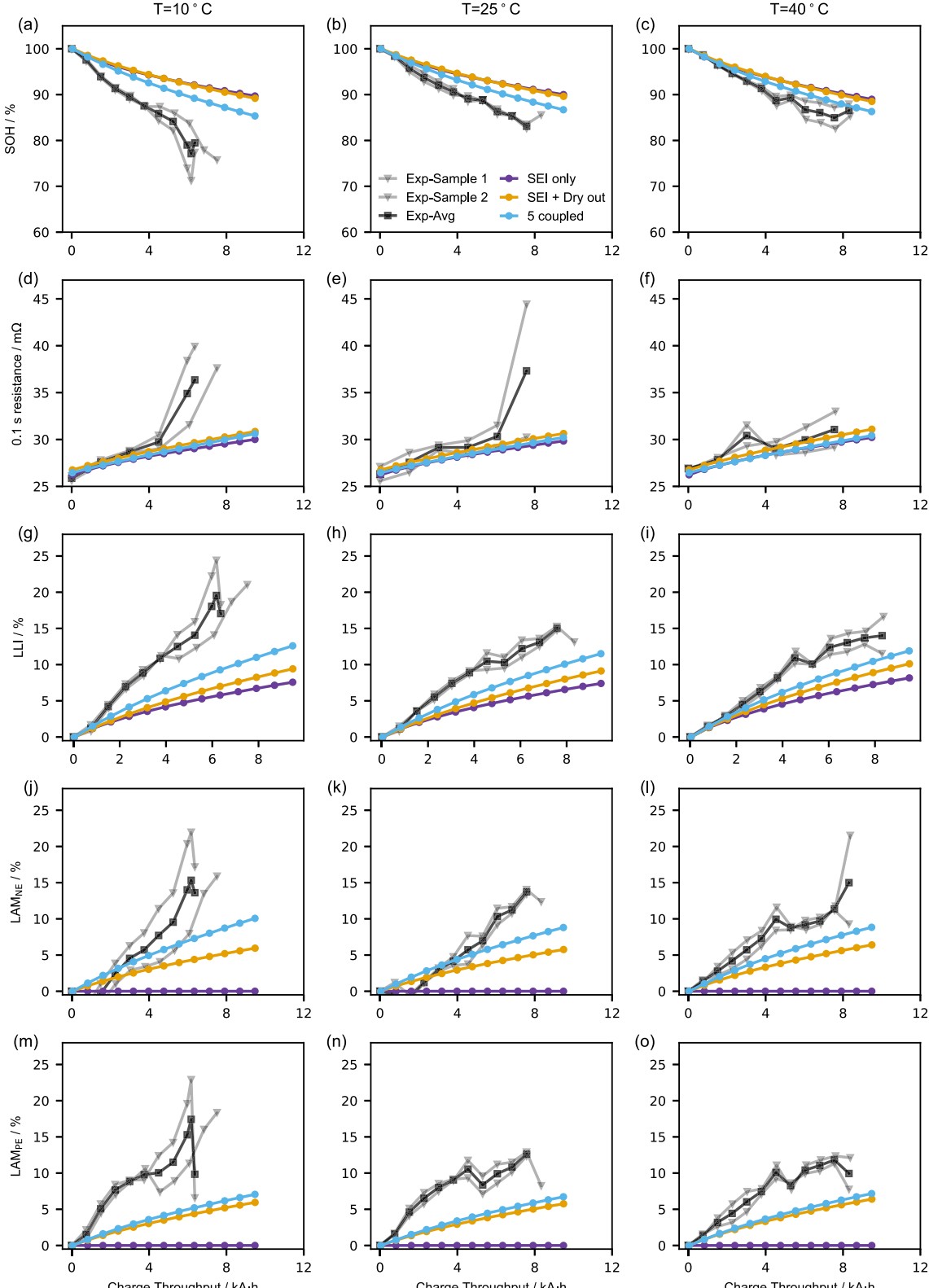

**Fig. 6 | Validation of the three models against Experiment 3.** State of health (SOH) (**a**–**c**), 0.1 s resistance (**d**–**f**), loss of lithium inventory (LLI) (**g**–**i**), loss of active material negative electrode (LAM$_{NE}$) (**j**–**l**), loss of active material positive electrode (LAM$_{PE}$) (**m**–**o**). Each row represents the same indices, and each column corresponds to the same temperature. Source data for this figure are provided as a Source Data file.

**Table 3 | Ageing parameters varied during parameter sweeping and their best fits for the three models**

| Ageing mechanism | Parameter | Unit | SEI only | SEI + Dry out | 5 coupled |
|---|---|---|---|---|---|
| SEI | Inner SEI lithium interstitial diffusivity ($D_{int}$) | $m^2/s$ | $2.36 \times 10^{-18}$ | $1.2 \times 10^{-18}$ | $9.81 \times 10^{-19}$ |
| | Inner or outer SEI partial molar volume ($\bar{V}_{SEI}^{inner}$ or $\bar{V}_{SEI}^{outer}$) | $m^3/mol$ | $4 \times 10^{-5}$ | $6.74 \times 10^{-5}$ | $5.22 \times 10^{-5}$ |
| | SEI growth activation energy ($E_{act}^{SEI}$) | J/mol | $1 \times 10^4$ | $1 \times 10^4$ | $5 \times 10^3$ |
| Lithium plating | Dead lithium decay constant ($\gamma_0$) | 1/s | – | – | $1 \times 10^{-7}$ |
| | Lithium plating kinetic rate constant ($k_{Li}$) | m/s | – | – | $1 \times 10^{-10}$ |
| LAM model | Positive electrode LAM constant proportional term ($\beta_{pos}$) | 1/s | – | – | $2.98 \times 10^{-18}$ |
| | Negative electrode LAM constant proportional term ($\beta_{neg}$) | 1/s | – | – | $2.84 \times 10^{-9}$ |
| Mechanical and cracking | Negative electrode cracking rate ($k_{cr}^{neg}$) | | – | – | $5.29 \times 10^{-25}$ |
| DFN model | Negative electrode diffusivity activation energy ($E_{act}^{D_{s,n}}$) | J/mol | $6 \times 10^4$ | $6 \times 10^4$ | $2 \times 10^4$ |

The ageing parameters are sourced from O'Kane et al.[11] but tuned to fit the following five indices of Experiment 2[32]: the commonly used indices of (1) capacity retention, (2) resistance, and those that are usually omitted, i.e., the DMs including (3) LLI, (4) LAM_NE, (5) LAM_PE. Details of how the DMs are calculated can be found in Supplementary Note 12. All these indices, except the resistance, are extracted from the 0.1 C discharge voltage curves during the RPTs. The resistance is obtained from the galvanostatic intermittent titration technique (GITT) curves during the RPTs. Details on how these indices are extracted from RPTs can be found in SI. To assist an overall evaluation, a weighted average of these five indices is created. In the absence of prior work to rely on, we propose the following weighting ratio:

$$\text{MPE}_{tot} = \frac{1}{2}\text{MPE}_{SOH} + \frac{1}{8}\text{MPE}_{Res} + \frac{1}{8}\text{MPE}_{LLI} + \frac{1}{8}\text{MPE}_{LAM_{NE}} + \frac{1}{8}\text{MPE}_{LAM_{PE}}$$

(1)

The higher weight given to SOH biases the chosen sub-model to be that which most accurately predicts SOH, as the most impactful cell property for end users. Of course, the choice of weights can and should be changed to align with the scenario considered. The best fits for the three degradation models are selected based on the smallest value of MPE_tot. Note that due to the large parameter space and the long time required to run the ageing model (4GB of memory, 2 CPUs, and 16 h on average for the 5 coupled models. For 10 °C it may take more than 72 h for some parameter combinations), the best fits presented in this work are obtained by brute force/trail-and-error. Specifically, 4 parameters in SEI-only model SEI + Dry out, and 9 parameters in 5 coupled are first given a range, then different combinations of these ageing parameters are generated using Latin Hypercube sampling[44], and several interactions are carried out. At the last stage of parameter tuning, to retrieve the temperature-dependent capacity loss, we have also conducted a parameter study on two key parameters, i.e., the SEI growth activation energy ($E_{act}^{SEI}$) and the negative electrode diffusivity activation energy ($E_{act}^{D_{s,n}}$), see Supplementary Note 13 and Supplementary Fig. 13. The final changeable ageing parameters are listed in Table 3. Those unchanged ageing parameters are listed in Supplementary Tables 8 and 9.

Because no optimisation methods are used in parameterisation, there may exist other parameter combinations which can achieve lower value of MPE_tot. However, the conclusions of this work are not affected by this limitation. A thorough sensitivity study and optimisation are in the scope of future works to further explore the predictive power of the model.

## Data availability
The experimental data used in this work are from Kirkaldy et al.[32]. The long-term simulation data, together with the used experimental data, can be found at Zenodo (https://zenodo.org/records/14995785) as ref. 45. Source data are provided in this paper.

## Code availability
The scripts and the resulting figures in this paper can be found at Zenodo (https://zenodo.org/records/14995785), as reference[45]. This script is licensed under a Creative Commons Attribution 4.0 International license. To view a copy of this licence, visit https://creativecommons.org/licenses/by/4.0/legalcode.

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

## Acknowledgements

The authors thank Dr. Kieran O'Regan for providing updated open-circuit potential data for the LG M50 cell, using the same methods as his published work. The authors are also grateful for financial support from the EPSRC Faraday Institution Multiscale Modelling project (EP/S003053/1, grant number FIRG059). Dr. Ruihe Li was funded as a PhD student by the China Scholarship Council (CSC) Imperial Scholarship.

## Author contributions

**R. Li**: Conceptualisation, Validation, Software, Formal analysis, Visualisation, Writing – Original Draft. **N.D. Kirkaldy**: Resources, Data Curation, Discussion. **F.F. Oehler**: Formal analysis, Discussion. **M. Marinescu**: Investigation, Writing – Review & Editing, Supervision. **G.J. Offer**: Conceptualisation, Writing – Review & Editing, Supervision, Funding acquisition, Project administration. **S.E.J. O'Kane**: Methodology, Writing – Review & Editing, Supervision.

## Competing interests

The authors declare no competing interests.
