## [Peer Review File · Nature Communications]

The importance of degradation mode analysis in parameterizing lifetime prediction models of lithium-ion battery degradation

Corresponding Author: Professor Gregory Offer

Version 0:

Reviewer comments:

Reviewer #1

(Remarks to the Author)

Accurate aging models are essential for predicting Remaining Useful Life, optimizing operating conditions, and improving manufacturing processes. The authors motivate their work with existing models being mostly validated only against capacity fade and resistance. They claim that parameterization that is limited to these is insufficient. Therefore, the authors use degradation modes. The paper examines five degradation mechanisms and combines them into three different models. By examining these combinations, the study aims to identify the most effective models for accurately predicting battery degradation.

In general, the authors address a very important issue, which is the identification and parameterization of degradation models. Their idea to use degradation modes in addition to capacity fade and resistance increase is straightforward. Their results are generally very interesting, and to the best of my knowledge this is really the most sophisticated work in this area. However, looking into the details of their study raised major concerns that need to be addressed before publication. The details are outlined below.

Major concerns:

- Validation of models: The authors argue that using the degradation modes approach helps to select and parameterize aging model. However, I have concerns about how the authors validate the model. A valid model captures the correct mechanisms and is therefore able to make predictions of capacity and resistance, i.e. the aspects that matter for the application, beyond the data used for parameterization. The true impact of an incorrect or poorly parameterized model may only become apparent when the model predicts new data, revealing the shortcomings of inaccurately predicted degradation modes. In other words, a better fit of the degradation modes is pointless if the performance in terms of predicting resistance increase and capacity fade remains unchanged. The main question, i.e., whether the degradation mode analysis significantly improves the prediction capability, is therefore not sufficiently addressed in this work.
- Parameter fit: The authors adjust the parameters of their degradation model to better fit the degradation data. Details of this process remain unclear. For example, the objective function of this fitting process is not clear. Is Equation 1 the objective function? In part one, they only fit against capacity and resistance. How is the objective function in this case defined? Do all simulations that are shown in the paper use the same set of parameters (Table 3) or are there different parameters identified for fitting against capacity and resistance? If they are different, it would be important to show them in the paper. Then they analyze the capability of the different models with respect to degradation modes. Obviously, the SEI-only model must fail in this regard. The comparison 5 degradation mechanism with SEI + dry out is not so clear. The performance is similar, only slightly better with 5 degradation mechanisms. It is important to note that 5 degradation mechanism model also has many more parameters that can be adjusted. Furthermore, are the identified parameters of the 5 degradation mechanism model unique? Or is there another parameter combination of the 5 degradation mechanism model that might have similar errors? It seems obvious to me that a model with more fitting parameters will give better results. This doesn't prove that the model is correct (see model validation). The negative electrode diffusion activation energy was adjusted from $6e4$ to $2e4$ for the 5 degradation mechanism model. But then the P2D model is no longer well parameterized at BOL, I suppose.
- Silicon: The authors seem to have neglected any silicon-specific degradation mechanisms. Isn't silicon usually the main driver of degradation? How is it justified that silicon degradation is neglected? Even assuming that only the implemented

degradation processes, e.g. SEI growth, occur in silicon anodes, the influence of voltage hysteresis could be significant. The authors point out that SEI growth is complex and it is very important to accurately predict the anode potential. Does the model account for voltage hysteresis? If not, how does that influence, the results of this study?

Minor

- Fig 2 only shows a low rate discharge. How about higher rates?
- Paragraph from L71-79 needs some references
- L96: meaning waysmeaningful way? Also what does that mean?
- L100 RPT not introduced
- L115 Light gray line does not exist in legend of Fig. 3
- L136 should be Fig.3(a)-(c), I suppose
- L145 Hard to understand for someone that does not know the model in detail.
- L153 Why is SOH (here capacity loss) the most impactful property. For some application that might rather be resistance.
- L159 Degradation mechanisms -> Degradation modes?
- L162 explains how DM is identified... should be addressed in method section or SI
- L175 what is meant by change shape? Peak size? Peak Location?
- L232 what is usually measured?
- L241 lump  lumped?
- L267 Reference for latin hypercube sampling is missing
- L272 exists- _> exist
- L277 "wired results"? What does that mean?

(Remarks on code availability)

Reviewer #2

(Remarks to the Author)

See attached pdf.

(Remarks on code availability)

No model was provided.

Version 1:

Reviewer comments:

Reviewer #1

(Remarks to the Author)

I would like to thank the authors for the effort they put into this comprehensive revision. I am mostly satisfied with the changes made, as they have addressed all my major concerns. I have two minor comments:

I have now learned that for all aging models the objective function was Equation 1. In this context, I still find the phrase "model fitting against voltage, capacity, and resistance" confusing. To me, the term "fitting" implies the process of adjusting the parameters, i.e., in this case, minimizing the difference for voltage, capacity, and resistance and thus ignoring the deviations for the degradation modes. However, this is not what you mean. Maybe you could adjust the wording. For example, "Simulation error for voltage, capacity and resistance".

The activation energy of diffusion in the negative electrode is included as a fitting parameter. Since the model is only parameterized for 25°C, I agree that the fitting will not affect this parameterization. However, it would change it if you included another temperature for the BOL parameterization. In my opinion, this is not an aging parameter. Why are you using it as an aging parameter when it is not part of the aging model? Could you explain your motivation for including this parameter?

(Remarks on code availability)

Reviewer #2

(Remarks to the Author)

Reviewer is satisfied with the response.

(Remarks on code availability)

Version 2:

Reviewer comments:

Reviewer #1

(Remarks to the Author)

I am satisfied with the changes that have been made. I recommend that the manuscript be accepted for publication.

(Remarks on code availability)

In this paper, the authors develop a P2D based degradation model validated with capacity fade, resistance increase. The novelty here is additional validation with 3 degradation mode data as well namely LLI, LAM NE and LAM PE. The degradation physics added into the model to get the metrics are SEI growth, electrolyte dryout, Li plating, and particle cracking. The number of degradation physics added is methodically varied from 1 to 2 to 5 to delineate that all degradation physics is needed to be added and not a single mechanism can be skipped to get the additional validation with LLI, LAM NE and LAM PE. Here are a few comments that need to be addressed:

Provide more details of experimental aging datasets for ease of understanding of the reader. The experimental dataset article is referenced (29), but it would be nice to have a schematic of the experimental protocol with highlighted portions that are being validated against in the Main or SI here itself.

Page 4, “The lump resistance is obtained by the voltage drop 0.1 s after the 12th pulse of a C/2 GITT discharge”: Please explain the nomenclature “lump resistance”. I have seen pulse resistance etc. terminology, not sure what “lump” stands for. Hybrid pulse power characterization tests are the general standardized methodology to characterize resistances. What is the rationale behind the method used in this article? Why is the time used only 0.1s (seems very short)?

Figure 3: What is the significance of the grey lines? There are 4 colors in the legend but 5 different colors on the plot? Are the grey lines the individual replicates of the experimental data? If so, point it out in the legend explicitly.

Figure 4: The legend is crossing into the trend lines for panel (a). Kindly ensure the figure quality is good all throughout.

Page 8, Discussion Related to MPEs: The results suggest that more physics leads to lower MPEs i.e. 5 coupled < SEI + Dry out < SEI only. More physics also leads to more adjustable parameters. Can the authors comment on how to move forward in the inclusion of physics such that we are capturing the degradation modes while also not just overfitting? Is there an optimal number of physics (like 5 coupled here) and fit parameters (like 9 in the 5 coupled model) beyond which any model will be able to accurately fit the degradation data like LLI, LAMNE, LAMPE?

Another question related to sub-model choice: Let's take the model for Li plating. Here a Li plating model that includes plating, striping and dead Li formation is taken from the author's previous article. There are plenty of Li plating models in literature for example: Ren *et al* 2018 *J. Electrochem. Soc.* **165** A2167, XG Yang et al. <https://doi.org/10.1016/j.jpowsour.2018.05.073>, Luders et al <https://doi.org/10.1016/j.jpowsour.2018.12.084>. Can the authors comment on how did they arrive at the choice of their Li plating model equations? Is there a way to decide that the plating model equations chosen are best representative of the corresponding physics? Will the results change significantly if, let's say, the authors choose the Li model equations from the different articles mentioned above?

Page 12, Model: The DFN model described in the SI uses incorrect equation for electrolyte mass conservation if the variation of transference number with concentration is explicitly considered like in the electrolyte properties figures below equation S62 in SI. There is a term which involves gradient of the transference number that is missing (see Equation 10 in Marc Doyle *et al* 1993 *J. Electrochem. Soc.* **140** 1526). How will the results be affected with the inclusion of this term correctly?

$$\varepsilon \frac{\partial c_e}{\partial t} = \nabla \cdot (D_e^{eff} \nabla c_e) - \frac{i_e \nabla t_+}{F} + \frac{j(1 - t_+)}{F}$$

In summary, this article provides detailed validation of degradation metrics other than the usual capacity fade and resistance using physics-based models. The conclusions like we need all five mechanisms to validate the degradation data is good, but not significantly novel. From the physics and model assumptions itself, SEI can only mimic LLI and the SEI + electrolyte dryout can only give equal LAMNE and LAMPE. This has been pointed out by the authors as well; so the 5 coupled model will be the best was a foregone conclusion from the very beginning of this research undertaking without the need to run a model. However, the article does a good job in actual model validation and giving quantitative numbers. It highlights the need to validate more degradation metrics like LLI, LAM for physics-based capacity fade models.

The point-by-point responses to referees are as follows, in which the contents in blue represent the original comments, the contents in black are our response, and the contents in red with quotation marks “ ” are the sentences in the revised manuscript.

Reviewer #1 (Remarks to the Author):

Accurate aging models are essential for predicting Remaining Useful Life, optimizing operating conditions, and improving manufacturing processes. The authors motivate their work with existing models being mostly validated only against capacity fade and resistance. They claim that parameterization that is limited to these is insufficient. Therefore, the authors use degradation modes. The paper examines five degradation mechanisms and combines them into three different models. By examining these combinations, the study aims to identify the most effective models for accurately predicting battery degradation.

In general, the authors address a very important issue, which is the identification and parameterization of degradation models. Their idea to use degradation modes in addition to capacity fade and resistance increase is straightforward. Their results are generally very interesting, and to the best of my knowledge this is really the most sophisticated work in this area. However, looking into the details of their study raised major concerns that need to be addressed before publication. The details are outlined below.

Major concerns:

#1-1 Validation of models: The authors argue that using the degradation modes approach helps to select and parameterize aging model. However, I have concerns about how the authors validate the model. A valid model captures the correct mechanisms and is therefore able to make predictions of capacity and resistance, i.e. the aspects that matter for the application, beyond the data used for parameterization. The true impact of an incorrect or poorly parametrized model may only become apparent when the model predicts new data, revealing the shortcomings of inaccurately predicted degradation modes. In other words, a better fit of the degradation modes is pointless if the performance in terms of predicting resistance increase and capacity fade remains unchanged. The main question, i.e., whether the degradation mode analysis significantly improves the prediction capability, is therefore not sufficiently addressed in this work.

Thanks so much for insightful suggestions. We have now added validation on “Experiment 3”, which was carried out together with “Experiment 2” used for parameterization in another paper in our group [1].

The results are shown below and added to the main text and SI. “The 5 couple model well predicts the all the five indices of “Experiment 3” at 40°C, leading to an MPE_{tot} of 10.58%. It only deviates from the experiment data at the last RPT where a small jump in SOH occurs, which can be ascribed to the

changes in LAM_{NE} and LAM_{PE} . The performance of the *5 couple* model at 25°C is worse than that at 40°C but still predict comparable values at the last RPT, while it completely failed at 10°C. This may be due to the relatively poor quality of the experiment data, as evidenced by the capacity recovery observed at 10°C and 25°C and the huge discrepancy between the three curves at 10°C, indicating the occurrence of degradation mechanisms that may not have been included in this work. Nonetheless, the *5 couple* model performs the best among the three models in capturing the degradation behaviours observed during cycling at high SOC ranges (85%~100% in “Experiment 3”), whereas the other two model underestimates the degradation in all indices in Fig. 6. The best performance of *5 couple* model on “Experiment 3” can be ascribed to its ability to predict extra stress driven LAM_{NE} at high SOC range (Fig. S18(f)).”

We acknowledge that the three model performs similarly on predicting resistance. We believe the most likely cause is because the DFN type model is a “lumped” model that can only consider “one point” in the electrode area. All these “points” in the electrode area can be regarded as connected in parallel. For quantities like capacity, total lithium inventory, active material mass, quantities of all these “points” can be added together. However, the total resistance of parallel connected networks follows $\frac{1}{R_{tot}} = \sum_i \frac{1}{R_i}$, where R_i is the resistance of the i point. Therefore, the effect of lump resistance can be captured better in a model that capture the in-plane inhomogeneity, such as an equivalent circuit network (ECN) model [2].

#1-2 Parameter fit: The authors adjust the parameters of their degradation model to better fit the degradation data. Details of this process remain unclear. For example, the objective function of this fitting process is not clear. Is Equation 1 the objective function? In part one, they only fit against capacity and resistance. How is the objective function in this case defined?

Yes, the objective function is our Eq. (1). Therefore, the two sections “**Model fitting against voltage, capacity, and resistance**” and “**Model fitting against degradation modes**” actually come from the same fitting process. The intention to separate these results into 2 sections is to highlight the storyline of this paper, i.e., how degradation models in most previous papers have been parameterised (mostly fitting to the voltage, capacity, and resistance) and why that is not enough.

However, we agree that such an arrangement is confusing. To make it clearer, we have moved the **Method** section forward and added a subsection **Parametrisation**.

#1-3 Do all simulations that are shown in the paper use the same set of parameters (Table 3) or are there different parameters identified for fitting against capacity and resistance? If they are different, it would be important to show them in the paper. Then they analyze the capability of the different models with respect to degradation modes. Obviously, the SEI-only model must fail in this regard.

Yes, all figures in the paper are produced with the input parameters in Table 3 except Fig. S11, where we explored the temperature dependency of the two key parameters $E_{\text{act}}^{\text{SEI}}$ and $E_{\text{act}}^{\text{D}_{s,n}}$. To avoid confusion, we have moved the parameter study on $E_{\text{act}}^{\text{SEI}}$ and $E_{\text{act}}^{\text{D}_{s,n}}$ into SI.

#1-4 The comparison 5 degradation mechanism with SEI + dry out is not so clear. The performance is similar, only slightly better with 5 degradation mechanisms. It is important to note that 5 degradation mechanism model also has many more parameters that can be adjusted.

We agree that in our original draft, the differences between the *5 coupled* and *SEI + dry out* is not clear enough. But now with our new validation on “Experiment 3” and the revised discussion, the differences will be much clearer.

For “Experiment 2”: “**The *5 coupled* models fit all DMs better than the *SEI + Dry out* model at all three temperatures (Error! Reference source not found.).** The improvement is significant for low temperature compared to higher temperatures, because the mechanical degradation in the *5 coupled* model is more severe at low temperature. regarding the overall judgement, the *5 coupled* model has overall lower MPE_{tot} (34.45%) for the three temperatures compared to that of the *SEI + Dry out* model (45.14%).”

For “Experiment 3”: “...the *5 couple* model performs the best among the three models in capturing the degradation behaviours observed during cycling at high SOC ranges (85%~100% in “Experiment 3”), whereas the other two model underestimates the degradation in all indices in **Error! Reference source not found.** The best performance of *5 couple* model on “Experiment 3” can be ascribed to its ability to predict extra stress driven LAM_{NE} at high SOC range (Fig. S18(f)).”

It is true that the *5 coupled* model has more adjustable parameters, and it is well known that having too many tuneable parameters is the “elephant in the room” for physics-based degradation models, i.e. with enough tuneable parameters you can fit an elephant. We have therefore been very careful throughout the paper to avoid claiming that the *5 coupled* model is correct, merely that we have found a parameter set with this physics that works. As a consequence, we decided to make the main storyline of the paper about proving the negative, and prove that models that do not use and/or capture how the degradation modes evolve over time are wrong. We believe proving the positive, that a particular model is ‘correct’ with a high level of significance, is something that the whole community is going to take many years to achieve.

#1-5 Furthermore, are the identified parameters of the 5 degradation mechanism model unique? Or is there another parameter combination of the 5 degradation mechanism model that might have similar errors? It seems obvious to me that a model with more fitting parameters will give better results. This doesn't prove that the model is correct (see model validation).

As acknowledged in our response above, it is very hard to prove that the identified parameter set in this work is unique, even with optimization method, as the parameter space is too huge. Therefore, we have only claimed the identified parameter combination in work to be “*a fit*” rather than “*the fit*”.

Conducting parameter optimization does bring more confidence. The main reasons that we have not applied any optimization methods is that the computational time required for one case is currently too long. To be specific, the *5 coupled* model in this work requires 4GB of memory, 2 CPUs, and 16 hours on average to finish one case (of ~one thousand charge/discharge cycles). However, the wall time for the jobs (for the definition of “jobs” in the context of HPC, the readers are referred to Chapter 5) submitted to the HPC system in Imperial College London is normally limited to 72 hours, with potential extension to 144 hours maximum. That means that a maximum number of only 9 cases can be finished within one single job. Many optimization methods require hundreds of interactions to find a local maximum/minimum for complicated problems and are normally designed to run continuously. Assuming 500 iterations are needed, the total time will be $500 \times 16 \div 24 \approx 333$ days, which is intolerable. What is worse, because we are using Casadi solver, the models are vulnerable to crash, which make optimization more difficult.

Such long simulation time is due to fact that we have chosen the “state-of-the art” physics-based modelling elements that are **publicly available** by the time this work is done. The complicated coupled models feature: (1) DFN model, rather than SPMe or SPM, to capture inhomogeneous degradation in the electrode thickness direction, (2) non-linear solid diffusivity to mimic the phase change behavior of graphite, (3) a lumped thermal model to capture temperature dependent ageing, (4) current sigmoid to capture the voltage hysteresis in silicon and graphite, (5) 5 degradation mechanisms, as discussed in the main text.

We agree with the reviewer that a model having more parameters can give better fits but still be wrong. However, that is exactly why we have introduced more indicators, i.e., DMs, to judge the model. As such, the more parameters brought by more degradation mechanisms are there not because we want more adjustable parameters to better fit the experimental data, but because they reveal the observable phenomena.

As mentioned above, we believe proving the positive, that a particular model is ‘correct’ with a high level of significance, is something that the whole community is going to take many years to achieve, and we are not claiming to have achieved in this paper.

#1-6 The negative electrode diffusion activation energy was adjusted from 6e4 to 2e4 for the 5 degradation mechanism model. But then the P2D model is no longer well parameterized at BOL, I suppose.

The BOL is only parameterised against 25°C only. Therefore, the activation energy does not matter in this case.

#1-7 Silicon: The authors seem to have neglected any silicon-specific degradation mechanisms. Isn't silicon usually the main driver of degradation? How is it justified that silicon degradation is neglected? Even assuming that only the implemented degradation processes, e.g. SEI growth, occur in silicon anodes, the influence of voltage hysteresis could be significant.

Thanks for the insightful comment. That is exactly our future direction. We want to highlight that during the time this work was carried out, there were no **publicly available** degradation models for the silicon/graphite composite electrode. However, our group was working on this in parallel. A recent paper [3] led by Dr. Bonkile, another member in our group, shows that the silicon degradation occurs mainly at lower SOC ranges (“Experiment 1” and “Experiment 5”), as indicated by Fig. A1.

To assist the reviewers’ understanding on the five experiments in Kirkaldy *et al.* [1], we have adapted Table A1 from [1].

Table A1. Summary of the cycle ageing study, showing conditions used and distribution of cells, adapted from [1].

Experiment	SOC window	Cycles per ageing set	Current (discharge/charge)	Temperature / °C	Number of cells
1	0~30%	258	0.3C/1C	10	3
				25	3
				40	3
2	70~85%	515	0.3C/1C	10	2
				25	2
				40	2
3	85~100%	515	0.3C/1C	10	3
				25	3
				40	3
4	0~100% (drive cycle)	78	0.3C/ noisy C	10	3
				25	2
				40	3
5	0~100%	78	0.3C/1C	10	3
				25	2
				40	3

To show this more practically, we have run the simulations and shown how poor the predictions of the model are for “Experiment 1” and “Experiment 5” where the silicon in the low SOC region is used, in Fig. S16 below or in SI. For these experiments all three models failed, and it is only the model with separate degradation for silicon and graphite in Dr. Bonkile’s paper that can reproduce those results.

Fig. A1. Degradation mode analysis for cell cycled at different SoC cycling ranges. Solid lines represent the simulation results, with shaded regions corresponding to the experiment results: (a) Loss of active material of the anode at 0-100% SoC range, (b) loss of active material of the silicon at 0-100% SoC range, and (c) loss of active material of the graphite at 0-100% SoC range as a function of charge throughput, (d) Loss of active material of the anode at 0-30% SoC range, (e) loss of active material of the silicon at 0-30% SoC range, and (f) loss of active material of the graphite at 0-30% SoC range as a function of charge throughput. Adapted from [3].

Fig. S16 Model validation against “Experiment 1” and “Experiment 5” at 40°C. All three model fail to predict both experiments.

#1-8 The authors point out that SEI growth is complex and it is very important to accurately predict the anode potential. Does the model account for voltage hysteresis? If not, how does that influence, the results of this study?

Yes, we have included a zero-order hysteresis model via the “current sigmoid” function, which can be found in the section **Zero-order hysteresis model** in SI.

The effect can be elucidated in Fig. A2. With the zero-order hysteresis model, the experimentally observed voltage hysteresis during 0.1C cycling can be well captured. As presented in Fig. A2, the zero-order hysteresis model will make the virtual cell spent less time at low voltage region, which will promote the degradation with the mechanisms considered in this work.

Fig. A2 Comparisons between the modelling and experimental 0.1C charge/discharge voltage curves (a) without and (b) with current sigmoid.

#1-9 Minor

#1-9-1 Fig 2 only shows a low rate discharge. How about higher rates?

I assume the reviewer is asking about the performance of the model in predicting the dynamic behaviour. We have also carried out validation against 1C and 2C GITT in Fig. S12 at BOL, which show good fits. The RMSEs for 1C and 2C GITT are 130 mV and 185 mV, respectively.

#1-9-2 Paragraph from L71-79 needs some references

Thank you so much for the kind suggestions. Related references have been added.

#1-9-3 L96: meaning waysmeaningful way? Also what does that mean?

Thank you for pointing that out. Yes, we mean “meaningful way” here.

#1-9-4 L100 RPT not introduced

Thanks so much for pointing that out. The full name of RPT has now been added when it first appears in the main text, with experiment details added in SI.

#1-9-5 L115 Light gray line does not exist in legend of Fig. 3

Sorry for the missing information. We have now added the following statements for Fig. 3 and other similar figures: “the grey lines represent the two test cells, and the black line is their average.”

#1-9-6 L136 should be Fig.3(a)-(c), I suppose

Thank you for pointing that out. We have corrected the typo as suggested.

#1-9-7 L145 Hard to understand for someone that does not know the model in detail.

Sorry for the inadequate discussion. We have now added more explanations: “The improvement is significant for low temperature compared to higher temperatures, as the mechanical degradation in the 5 coupled model is more severe at low temperatures. This is because low temperatures lead to lower solid diffusivities of the electrodes (Eq. (S64)) and therefore larger lithium concentration gradients in the particles and ultimately more stress-driven LAM (Eq. (S20) and Fig. S9).”

#1-9-8 L153 Why is SOH (here capacity loss) the most impactful property. For some application that might rather be resistance.

We agree with the reviewers that resistance may be more important to track in some scenarios. However, as far as we know, SOH (representing capacity in this work) is the most frequent indices that the academic community fit to when it comes to degradation modelling of LIBs. Moreover, we believe that the resistance of aged cells may be better predicted with a model considering higher dimensions [2] rather than the one geometric dimension, DFN style model. More discussion on this can be found in our response to the first comment.

#1-9-9 L159 Degradation mechanisms -> Degradation modes?

Thank you for pointing that out this typo. We have corrected it as suggested.

#1-9-10 L162 explains how DM is identified... should be addressed in method section or SI

Thank you for the important advice. We initially omitted that because DM analysis is a well-established method that has been widely used in the battery community. However, we appreciate the reviewer in that more details will make our work more readable. We have now added a new section **Calculation of DMs** in SI for that.

#1-9-11 L175 what is meant by change shape? Peak size? Peak Location?

Thank you for the comment. When visually describing differential voltage analysis, it is normal to comment on how the peaks change shape, size and location. However, what that means is not obvious unless a reader is familiar with the technique and discussion of this type. To acknowledge this and help the reader we have included the following and a reference to a recent paper that explains this well and references all the other key work in this area. “For the reader unfamiliar with these curves, Weng *et al.* [4] provide a good overview and description of how they change in response to different degradation modes.”

#1-9-12 L232 what is usually measured?

Sorry for the inadequate discussion. Here we mean that $LAM_{NE}=LAM_{PE}$ has not been observed in previous experiments. We have rephrased as follows. “...although the dry-out sub-model describes the evolution of LAM, the modelled mechanism implicitly predicts $LAM_{NE}=LAM_{PE}$, which, to our best knowledge, is not observed in previous experiment works^{33, 37-39}.”

#1-9-13 L241 lump  lumped?

Thank you for pointing that out this typo. We have corrected this in the whole draft as suggested.

#1-9-14 L267 Reference for latin hypercube sampling is missing

Thank you for pointing that out. We have cited the original work of this method: McKay, M. D., Beckman, R. J., & Conover, W. J. (1979). A comparison of three methods for selecting values of input variables in the analysis of output from a computer code. *Technometrics*, 21(2), 239-245. doi:10.2307/1268522

#1-9-15 L272 exists- _> exist

Thank you for pointing that out this typo. We have corrected it as suggested.

#1-9-16 L277 “wired results”? What does that mean?

Sorry for the inaccurate wording. Here we refer to the “**capacity recovery observed at 10°C and 25°C and the huge discrepancy between the three curves at 10°C**”.

Reviewer #2 (Remarks to the Author):

See attached pdf.

Reviewer #2 (Remarks on code availability):

#2-1 No model was provided.

Thank you so much for the kind suggestions. We have now made the script and long-term simulation data public and add a new section **Data and code availability** in the main text to guide the readers to there:

“The experimental data used in this work are from Kirkaldy *et al.* [1]. The long-term simulation data, together with the used experimental data can be found at Zenodo (<https://zenodo.org/records/13400193>). Code used in this paper are available from GitHub at (https://github.com/ImperialCollegeLondon/PyBaMM-ESE-Public/tree/GEM-2_NC/Reproduce_Li2024).”

In this paper, the authors develop a P2D based degradation model validated with capacity fade, resistance increase. The novelty here is additional validation with 3 degradation mode data as well namely LLI, LAM NE and LAM PE. The degradation physics added into the model to get the metrics are SEI growth, electrolyte dryout, Li plating, and particle cracking. The number of degradation physics added is methodically varied from 1 to 2 to 5 to delineate that all degradation physics is needed to be added and not a single mechanism can be skipped to get the additional validation with LLI, LAM NE and LAM PE.

Here are a few comments that need to be addressed:

#2-2 Provide more details of experimental aging datasets for ease of understanding of the reader. The experimental dataset article is referenced (29), but it would be nice to have a schematic of the experimental protocol with highlighted portions that are being validated against in the Main or SI here itself.

Thank you so much for the kind suggestions. That will definitely make this work more readable. We have added a subsection **Ageing datasets** under the **Method** section in the main text, and also a section **Experiment details** in the SI to provide more information.

#2-3 Page 4, “The lump resistance is obtained by the voltage drop 0.1 s after the 12th pulse of a C/2 GITT discharge”: Please explain the nomenclature “lump resistance”. I have seen pulse resistance etc. terminology, not sure what “lump” stands for. Hybrid pulse power characterization tests are the general standardized methodology to characterize resistances. What is the rationale behind the method used in this article? Why is the time used only 0.1s (seems very short)?

Sorry for the misleading wording, we have change that to 0.1 s resistance to be more specific. The time selected to be 0.1 second is to follow what has been done in the original experiment paper [1]. For such short time of 0.1 second (therefore a frequency of 10 Hz), the obtained resistance is mainly attributed to ohmic resistances

#2-4 Figure 3: What is the significance of the grey lines? There are 4 colors in the legend but 5 different colors on the plot? Are the grey lines the individual replicates of the experimental data? If so, point it out in the legend explicitly.

Sorry for the missing information. We have now added the following statements for Fig. 3 and other similar figures: “**the grey lines represent the two test cells, and the black line is their average.**”

#2-5 Figure 4: The legend is crossing into the trend lines for panel (a). Kindly ensure the figure quality is good all throughout.

Thank you for pointing that out. We have corrected this in the revised draft.

#2-6 Page 8, Discussion Related to MPEs: The results suggest that more physics leads to lower MPEs i.e. 5 coupled < SEI + Dry out < SEI only. More physics also leads to more adjustable parameters. Can the authors comment on how to move forward in the inclusion of physics such that we are capturing the degradation modes while also not just overfitting? Is there an optimal number of physics (like 5 coupled here) and fit parameters (like 9 in the 5 coupled model) beyond which any model will be able to accurately fit the degradation data like LLI, LAMNE, LAMPE?

Thank you so much for raising this important question. This question is what the whole battery community needs to address carefully to really make the models beneficial to the battery industry.

The short answer is, parameterising each degradation sub-model against a separate experiment dataset that mainly triggers that specific degradation mechanisms, then couple them together. This strategy requires a group of modellers and experimentalists working closely together, to produce the state-of-the-art models and high quality experiment data, and maybe more importantly, make them all public.

That is exactly what is happening in our group. On the one hand, we have a team of modellers who focus on different degradation mechanisms involving silicon composite electrodes [5-9], electrolyte consumption [10], lithium plating [11], cathode degradation [12, 13]. On the other hand, we have a group of experimentalists who focus on conducting well-design experiments and collect high-quality, open-source data [1, 14].

Our first paper on coupled degradation model with four degradation mechanisms is published in 2022 [15]. However, that paper, though reproducing five degradation paths based on the usage conditions, lacks validation. In this work, we have coupled one more degradation mechanism, the electrolyte dry-out, and taken the first step in parameterising and validating such complicated models, which is a further advancement.

However, we fully acknowledge that the work is not finished, with more degradation models being developed and to be coupled (such as silicon composite electrodes [5-9] and cathode degradation [12, 13]); and more experiment data being collected [14].

Clearly attempting to fit a complicated model with so many fitting parameters to a single dataset will almost certainly lead to overfitting. However, forcing each sub-model to fit experiments carefully designed to accelerate a sub-set of the mechanisms, and then putting them together, and validating should result in models that are parameterisable and validatable, and ultimately trustworthy. Therefore, we believe we and other groups around the world are on the right track.

We haven't included this discussion in the paper, as it is more philosophical rather than specific to this paper. However, we regularly get asked this type of question, and give this type of answer at conferences.

Fig. A3 Bottom-up philosophy of physics-based degradation models in our Electrochemical Science and Engineering (ESE) group.

#2-7 Another question related to sub-model choice: Let's take the model for Li plating. Here a Li plating model that includes plating, stripping and dead Li formation is taken from the author's previous article. There are plenty of Li plating models in literature for example: Ren et al 2018 J. Electrochem. Soc. 165 A2167, XG Yang et al. <https://doi.org/10.1016/j.jpowsour.2018.05.073>, Luders et al <https://doi.org/10.1016/j.jpowsour.2018.12.084>. Can the authors comment on how did they arrive at the choice of their Li plating model equations? Is there a way to decide that the plating model equations chosen are best representative of the corresponding physics? Will the results change significantly if, let's say, the authors choose the Li model equations from the different articles mentioned above?

Thanks so much for the important question. We have chosen the “state-of-the art” sub-models for each mechanism that are **publicly available in PyBaMM** by the time this work is done, which is largely based upon our previous work [15] except for the SEI sub-model. For the SEI sub-model, we have substituted the solvent diffusion limited model used in [15] with the interstitial diffusion limited model, as they are proven to better reproduce the SOC dependency and time dependency of SEI growth observed in experiments [16, 17]. For other sub-models inherited from our previous work [15], the justifications are provided there, see Section **2 Degradation mechanisms** in [15]. Specially, there are two paragraphs in [15] that discuss lithium plating models, which includes all three papers the reviewer mentioned:

“The first model of Li plating and stripping on graphite was reported by Arora, Doyle and White,²¹ who used a Butler–Volmer equation for plating/stripping. However, there is no dependence on the amount of plated Li in their model, causing it to predict negative values of the plated Li when all Li is stripped. This means the model can only be used for charge and not for discharge. Despite this limitation, the model received experimental validation from Ge et al.²²

The three-way interaction between Li plating, stripping and SEI formation has resulted in a range of models in recent years. Yang et al.²³ used a simplified version of Safari et al.'s¹⁶ model of SEI growth and a simple Tafel equation for irreversible Li plating. Their follow-up paper²⁴ introduced an updated Butler–Volmer equation including dependence on the amount of plated Li, taken from work on lithium metal batteries.²⁵ Ren et al.¹⁹ and von Lüders et al.²⁶ used a different approach, multiplying Arora, Doyle and White's original equation by a function that becomes zero when all the lithium is stripped, preventing the plated Li from going negative. Zhao et al.²⁰ let a preset fraction of plated Li turn into irreversible SEI instantaneously upon plating. Keil and Jossen¹¹ made this fraction time-dependent to achieve an excellent fit to experimental data. However, no models exist where plated Li gradually turns into SEI over time. A first step towards such a model is proposed in this work.” – adapted from [15].

Further discussion on the effect of the different model choices for all the different degradation mechanisms is not possible within the word limit of this paper. To answer the specific question on would it change the result, despite that we cannot compare them directly, we believe the results presented in this work will not be affected significantly by the choice of lithium plating models, as lithium plating only plays a small role in the overall degradation, see Fig. S18 in SI and below.

Finally, regarding the question “*Is there a way to decide that the plating model equations chosen are best representative of the corresponding physics?*”, we think that applies to other degradation models as well. Please refer to our response to the last question (#2-6) which involves our group’s long-term strategies on modelling.

Fig. S18 Contributions of different degradation mechanisms on DMs for the three models.

#2-8 Page 12, Model: The DFN model described in the SI uses incorrect equation for electrolyte mass conservation if the variation of transference number with concentration is explicitly considered like in the electrolyte properties figures below equation S62 in SI. There is a term which involves gradient of the transference number that is missing (see Equation 10 in Marc Doyle et al 1993 J. Electrochem. Soc. 140 1526). How will the results be affected with the inclusion of this term correctly?

$$\varepsilon \frac{\partial c_e}{\partial t} = \nabla \cdot (D_e^{eff} \nabla c_e) - \frac{i_e \nabla t_+}{F} + \frac{j(1 - t_+)}{F}$$

Thank you so much for pointing this out. This is our typo, which we correct now. We have used the built-in DFN model implemented in PyBaMM [18], which actually uses the correct form of the DFN model as suggested by the reviewer.

In summary, this article provides detailed validation of degradation metrics other than the usual capacity fade and resistance using physics-based models. The conclusions like we need all five mechanisms to validate the degradation data is good, but not significantly novel. From the physics and model assumptions itself, SEI can only mimic LLI and the SEI + electrolyte dryout can only give equal LAMNE and LAMPE. This has been pointed out by the authors as well; so the 5 coupled model will be the best was a foregone conclusion from the very beginning of this research undertaking without the need to run a model. However, the article does a good job in actual model validation and giving quantitative numbers. It highlights the need to validate more degradation metrics like LLI, LAM for physics-based capacity fade models.

- [1] N. Kirkaldy, M.A. Samieian, G.J. Offer, M. Marinescu, Y. Patel, *Journal of Power Sources*, 603 (2024).
- [2] S. Li, C. Zhang, Y. Zhao, G.J. Offer, M. Marinescu, *Communications Engineering*, 2 (2023).
- [3] M.P. Bonkile, Y. Jiang, N. Kirkaldy, V. Sulzer, R. Timms, H. Wang, G. Offer, B. Wu, *Journal of Power Sources*, 606 (2024).
- [4] A.D. Weng, J.B. Siegel, A. Stefanopoulou, *Frontiers in Energy Research*, 11 (2023) 18.
- [5] Y. Jiang, G. Offer, J. Jiang, M. Marinescu, H. Wang, *Journal of The Electrochemical Society*, 167 (2020).
- [6] Y. Jiang, Z. Niu, G. Offer, J. Xuan, H. Wang, *Journal of The Electrochemical Society*, 169 (2022).
- [7] W. Ai, N. Kirkaldy, Y. Jiang, G. Offer, H. Wang, B. Wu, *Journal of Power Sources*, 527 (2022).
- [8] H.J. Ruan, J.Y. Chen, W.L. Ai, B. Wu, *Energy and Ai*, 9 (2022) 13.
- [9] M.P. Bonkile, Y. Jiang, N. Kirkaldy, V. Sulzer, R. Timms, H.Z. Wang, G. Offer, B.L.Y. Wu, *Journal of Energy Storage*, 73 (2023) 10.
- [10] R. Li, S. O'Kane, M. Marinescu, G.J. Offer, *Journal of the Electrochemical Society*, 169 (2022) 14.
- [11] S.E.J. O'Kane, I.D. Campbell, M.W.J. Marzook, G.J. Offer, M. Marinescu, *Journal of the Electrochemical Society*, 167 (2020) 11.
- [12] A. Ghosh, J.M. Foster, G. Offer, M. Marinescu, *Journal of the Electrochemical Society*, 168 (2021) 14.
- [13] M.Z. Zhuo, G. Offer, M. Marinescu, *Journal of Power Sources*, 556 (2023) 16.
- [14] I.C.L.E. Group, in, 2024.
- [15] S.E.J. O'Kane, W.L. Ai, G. Madabattula, D. Alonso-Alvarez, R. Timms, V. Sulzer, J.S. Edge, B. Wu, G.J. Offer, M. Marinescu, *Physical Chemistry Chemical Physics*, 24 (2022) 7909-7922.
- [16] F. Single, A. Latz, B. Horstmann, *Chemsuschem*, 11 (2018) 1950-1955.
- [17] L. von Kolzenberg, A. Latz, B. Horstmann, *Chemsuschem*, 13 (2020) 3901-3910.
- [18] V. Sulzer, S.G. Marquis, R. Timms, M. Robinson, S.J. Chapman, *Journal of Open Research Software*, 9 (2021).

The point-by-point responses to referees are as follows, in which the contents in blue represent the original comments, the contents in black are our response, and the contents in red with quotation marks “ ” are the sentences in the revised manuscript.

Reviewer #1 (Remarks to the Author):

REVIEWER COMMENTS

I would like to thank the authors for the effort they put into this comprehensive revision. I am mostly satisfied with the changes made, as they have addressed all my major concerns. I have two minor comments:

I have now learned that for all aging models the objective function was Equation 1. In this context, I still find the phrase "model fitting against voltage, capacity, and resistance" confusing. To me, the term "fitting" implies the process of adjusting the parameters, i.e., in this case, minimizing the difference for voltage, capacity, and resistance and thus ignoring the deviations for the degradation modes. However, this is not what you mean. Maybe you could adjust the wording. For example, "Simulation error for voltage, capacity and resistance".

Sorry for the inappropriate phrase. Thank you so much for the kind suggestion. We totally agreed with the reviewer and have changed the phrase “Model fitting against ...” to **"Simulation error for ..."**. We have also changed other places where “fitting” is inappropriate to some other more appropriate, as highlighted a yellow in the revised manuscript.

The activation energy of diffusion in the negative electrode is included as a fitting parameter. Since the model is only parameterized for 25°C, I agree that the fitting will not affect this parameterization. However, it would change it if you included another temperature for the BOL parameterization. In my opinion, this is not an aging parameter. Why are you using it as an aging parameter when it is not part of the aging model? Could you explain your motivation for including this parameter?

Thanks so much for the insightful comment. We have adjusted the activation energy of diffusion in the negative electrode ($E_{act}^{D_{s,n}}$) mainly for the following three reasons: (1) it is given as a range in O'Regan et al. [1], which means the normal BOL parameterization tests is NOT good enough to determine its exact value. Therefore, it can / should be further identified by ageing datasets. (2) It affects the degradation behavior significantly, i.e., **“the higher the $E_{act}^{D_{s,n}}$, the lower the diffusivity and the lower the negative electrode potential during charge, ultimately leading to the higher the SEI growth rate.”** **“ $E_{act}^{D_{s,n}}$ also affects the stress driven LAM_{NE} , i.e., a higher value of $E_{act}^{D_{s,n}}$ gives a lower solid diffusivity**

under low temperatures, and therefore a higher stress-driven LAM.”

In this sense, we think that it may be difficult to clearly distinguish “aging parameter” and “BOL parameters”, as they are coupled together through the physics-based model (the mathematical equation), especially in terms of SEI.

Reference

[1] O'Regan, K.; Brosa Planella, F.; Widanage, W. D.; Kendrick, E. Thermal-electrochemical parameters of a high energy lithium-ion cylindrical battery. *Electrochimica Acta* **2022**, *425*. DOI: 10.1016/j.electacta.2022.140700.